# Diphoterine for Chemical Burns of the Skin: A Systematic Review

Felicia Dinesen [1],*, Pernille Pape [1], Martin Risom Vestergaard [1] and Lars Simon Rasmussen [1,2]

1   Department of Anaesthesia, Centre of Head and Orthopaedics, Section 6011 Rigshospitalet, Copenhagen University Hospital, DK-2100 Copenhagen, Denmark
2   Department of Clinical Medicine, University of Copenhagen, DK-2200 Copenhagen, Denmark
*   Correspondence: felicia.dinesen.01@regionh.dk

**Abstract:** The incidence of chemical burns appears to be increasing. Diphoterine is an amphoteric, chelating, polyvalent solution used for the decontamination of chemical splashes. In this systematic review, we aimed to assess the effect of diphoterine on chemical burns compared with water or no treatment. The primary endpoint was the depth of burn, and secondary outcomes included pain, duration of hospitalization, time to return to work, need for surgery, pH, and complications. PubMed, Embase, Cochrane Library, Web of Science, and Google Scholar were systematically searched using the terms "Diphoterine", "Previn", and ""Amphoteric solution" AND "burn"". A total of nine studies were included. One study evaluated the depth of chemical burns and found no difference between the diphoterine group and the control group. Four studies reported on pain, three of which found a more pronounced decrease in pain when using diphoterine compared to the control groups. Two studies found a significant neutralization of pH when using diphoterine. No differences were found for the remaining endpoints. Based on the very low certainty of evidence, this systematic review reports no observed difference between diphoterine and water or no treatment on the depth of a chemical burn. Diphoterine appeared to be associated with less pain and to have a neutralizing effect.

**Keywords:** chemical burn; skin burns; chemical splash

## 1. Introduction

Chemical burns represent a small proportion of burn injuries, but some studies have suggested an increase in the incidence of chemical burns [1]. The incidence varies from 3% [2] to 7.9% [1] of total burn injuries, and incidents now more often occur in a domestic setting than an industrial setting, with men being exposed to chemical burns more often than women [1].

Chemical burns differ from thermal burns in the way the damage spreads; therefore, treatment is different. In thermal burns, damage spreads due to thermal reaction. Acid burns cause proteins to denature and create a coagula, resulting in coagulative necrosis, while alkali burns cause proteins to denature and create a liquefactive necrosis, which, over time, allows deeper penetration into the skin. A quick elimination of the residual chemical product on the skin is key [3,4]. Due to the number of different chemical burning agents, there is variation in the severity of injury and a lack of evidence to recommend a specific treatment. Whether due to an acid or alkali agent, it is recommended to rinse a chemically burned area as soon as possible. If no rinsing solution is at hand, rinsing with water should be chosen. Water will primarily remove the burning agent from the skin and thereby prevent further damage [5].

Diphoterine (Prevor) is an amphoteric, chelating, polyvalent, slightly hypertonic solution made for the decontamination of chemical splashes. It is suitable for both acid and alkali burns, and its multiple binding sites provide it with the ability to effectively and quickly ameliorate tissue damage without causing an exothermic reaction. It should be

applied as soon as possible, but if it is not available immediately, it is recommended to first rinse with water, followed by rinsing with diphoterine [6]. This may lead to less pain, faster irrigation of pH, and possibly fewer days off work [7,8]. Previn is the German equivalent of diphoterine.

This systematic review sought to evaluate the current evidence on diphoterine used on chemical burns of the skin in humans.

We aimed to evaluate whether diphoterine is associated with a decrease in the depth of a chemical burn compared to rinsing only with water or no treatment.

Furthermore, we wanted to evaluate the effect of diphoterine compared with only rinsing with water or no treatment when looking at pain, time of hospitalization, time to return to work, and the need for surgery.

The neutralizing effect on chemical burns when looking at pH and complications associated with the use of diphoterine were also assessed.

## 2. Materials and Methods

This systematic review is reported in accordance with the Preferred Reporting Items for Systematic Reviews and Meta-Analysis (PRISMA) guidelines [9]. A PRISMA checklist has been published separately as a Supplementary File (File S1). The protocol for this study was registered in the PROSPERO database of systematic reviews on 16 March 2021 and published on 16 April 2021 (registration number: CRD42021243156) [10].

### 2.1. Eligibility Criteria

The research question was developed following the PICO (population, intervention, comparison, and outcome) framework. We identified studies recruiting humans exposed to a chemical splash on the skin of acid, alkali, liquid, or gas (population) that was rinsed with diphoterine (intervention) or water or that received no treatment (comparator). Only in vivo studies were included. Studies reporting on the depth of a chemical burn after treatment (primary endpoint), pain, duration of hospitalization, time to return to work, need for surgery, pH, and complications associated with the use of diphoterine (secondary endpoints) were included.

We excluded studies on diphoterine without a comparator. However, studies on complications associated with diphoterine were included even if no comparator was part of the study.

Reviews, systematic reviews, case-controls, case reports, case-series, letters, and expert opinions were also excluded.

Both peer-reviewed and non-peer-reviewed papers were included.

Authors of included abstracts with missing articles were contacted in order to obtain parts of the work or the full article. For contacts with missing responses, the abstracts were excluded. This was also the procedure regarding included conference abstracts.

There were no restrictions regarding publication year.

Furthermore, there were no restrictions regarding language. Full-text reading, data extraction, and the risk-of-bias assessment were completed in cooperation with a healthcare professional with skill in the given language.

### 2.2. Information Sources

One author, F.D., conducted a literature search in PubMed, Embase, the Cochrane Library, and Web of Science on 22 March 2021 using the terms "Diphoterine", "Previn", and "Amphoteric solution" AND "burn". Additionally, Google Scholar was searched on the same premises in order to broaden the search. PROSPERO and ClinicalTrial.gov (accessed on 20 March 2021) were consulted for similar ongoing studies. Reference lists from included studies were screened for additional relevant studies.

The search was updated on 8 July 2021 and again on 5 August 2022, and we found no new relevant studies.

### 2.3. Study Selection

Two authors, F.D. and P.P., independently screened the titles and abstracts of records found in the primary search by using the platform Covidence. Discrepancies were resolved by discussion until agreement was reached. The same two reviewers evaluated the full texts of the relevant reports. Again, discrepancies were resolved by discussion until consensus was obtained. Interrater reliability was calculated using Cohen's kappa statistic [11]. Reference lists from the included studies were screened by two authors, and reference lists from reviews found in the initial search were screened by one author.

### 2.4. Data Collection Process and Data Items

The data were extracted by two authors, F.D. and P.P., independently. This was completed using Covidence. Discrepancies were resolved by discussion until agreement was reached. The study details included location, study period, funding, and conflicts of interest. The data on methods included study design, eligibility criteria, exact intervention, and exact comparison. The data on outcomes included numbers of participants in each group, summary data in each group, between-group estimates, and adverse events.

### 2.5. Risk of Bias Assessment

Two authors, F.D. and P.P., assessed the risk of bias independently. Cochrane's tool was used for assessing the risk of bias in randomized trials. With that tool, the risk of bias can be judged as low, as presenting some concerns, or as high based on 5 domains covering the randomization process, including random sequence generation and allocation concealment; deviations from the intended interventions, including blinding, missing outcome data, and the measurement of the outcome; and the selection of the reported result.

For the non-randomized studies, quality was assessed using the Newcastle-Ottawa Scale (NOS). With that tool, a study was rewarded with stars from 0 to 9, with 9 stars representing a low risk of bias and 0 stars representing a high risk of bias. The scale contains 3 main domains covering selection (maximum of 4 stars), comparability (maximum of 2 stars), and outcome (maximum of 3 stars).

Discrepancies were resolved by discussion until agreement was reached.

To address possible selective reporting within studies, one author went through the included studies to compare the outcomes in the methods sections to those presented in the results. The outcomes in the methods sections were used for comparison due to a lack of published protocols.

### 2.6. Synthesis Methods

For each study, a summary statistic was done. The studies were described in a combined table presenting the studies that addressed the primary outcome at the top, followed by the studies addressing the secondary outcomes, in the following order: pain, duration of hospitalization, time to return to work, the need for surgery, pH, and complications associated with the use of diphoterine. Where the numbers of participants, means, and standard deviations were reported, 95% confidence intervals were calculated.

### 2.7. Certainty Assessment

The strength and quality of the body of evidence was assessed according to GRADE. Observational studies started at "low" by default. If the risk of bias, inconsistency, indirectness, or imprecision was rated as "serious", the quality of evidence was downgraded. The quality of evidence was upgraded if there was a large effect.

## 3. Results

Our primary search of the literature identified 817 records. After the removal of duplicates, 599 records were assessed for inclusion by title and abstract screening. Of these, 55 reports were included for full-text screening, and of these, 46 reports were excluded. For 13 reports, no full text was available, and the authors of the reports were contacted by

email or phone in an attempt to obtain data or a full-text article, but it was not possible to obtain the data, and they were excluded as well. Further, additional duplicates were found when full-text reading was completed due to variations in title and abstract in the records, leaving nine studies for inclusion in our review [8,12–19] (Figure 1, Table 1). The kappa coefficient was 0.63 (0.51–0.74) for title and abstract screening and 0.93 (0.79–1.00) for full-text screening, indicating substantial agreement and almost perfect agreement, respectively.

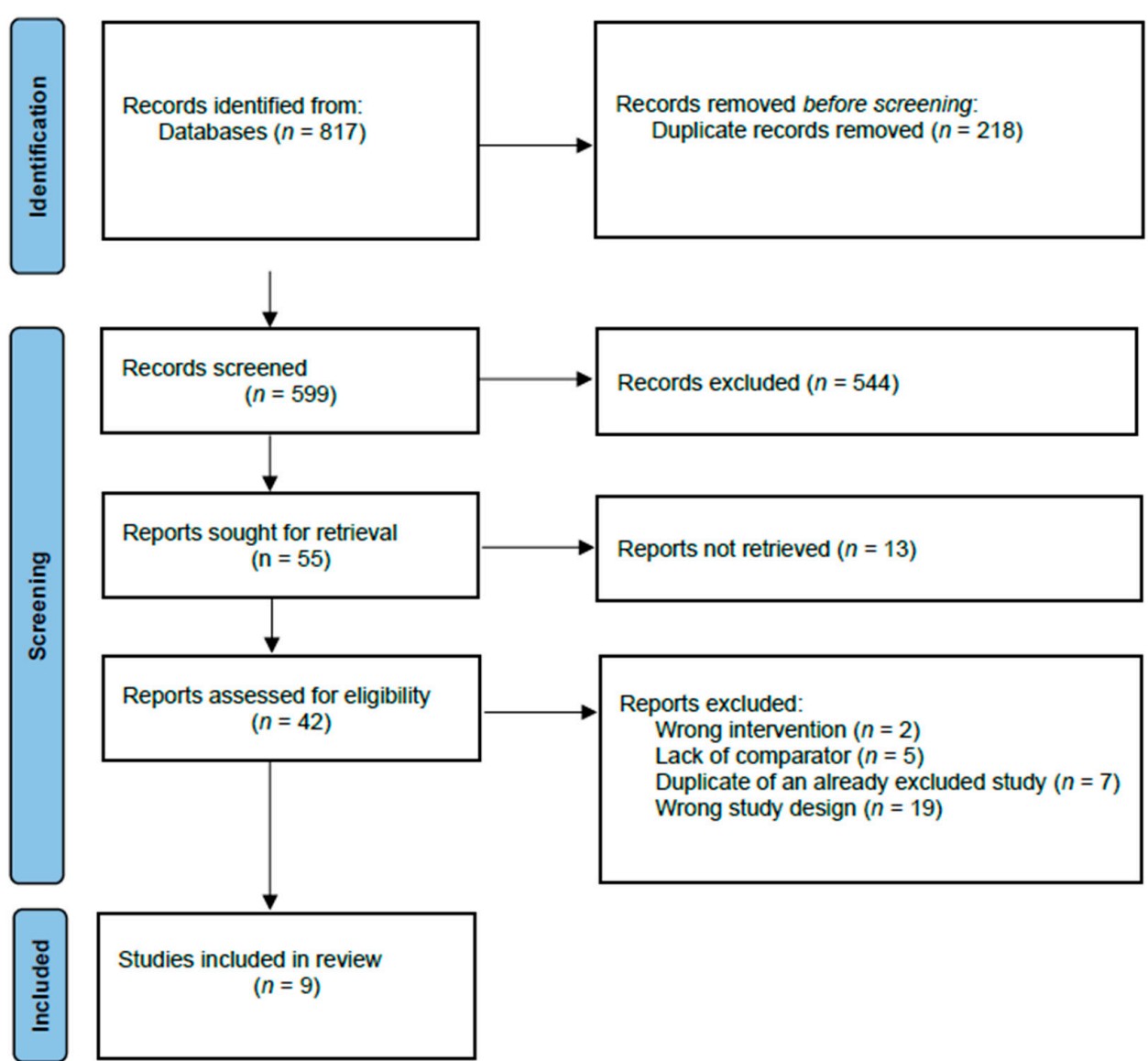

**Figure 1.** Identification of studies.

**Table 1.** Characteristics of the studies comparing Diphoterine with water or no treatment for chemical burns of the skin.

| Author | Location and Study Period | Method | Study Group | Injury Type | Intervention | Control | Burn Area (TBSA [†]) | Assessment |
|---|---|---|---|---|---|---|---|---|
| Bvrar, 2016 [13] | Slovenia, no study year reported | Randomized controlled trial, no blinding | 22 police officers | Exposure to CS [¤] gas | Two groups. Preexposure group, who sprayed face and eyes with 200 mL diphoterine solution just before CS exposure (eight officers) Postexposure group, who sprayed face and open eyes with 200 mL diphoterine solution immediately after CS exposure (eight officers) | No treatment (six officers) | Data not available | Pain, NRS 0–10 (mean, 95% CI), time to return to work (min:sec) |
| Cavallini, 2010 [18] | Italy, no study year reported | Non-randomized controlled trial, no blinding | 25 patients | Chemical peel on both forearms using 70% glycolic acid solution for 5 min | Rinsing right forearm with diphoterine | Washing left forearm with water for 1 min and afterwards rinsing with diphoterine | Data not available | pH (mean) litmus paper |
| Donoghue, 2010 [17] | Australia, May 2005 to April 2006 and May 2007 to April 2008 (24 months) | Retrospective observational study of records from three alumina refineries | 266 workers | Alkali chemical splashes to the skin | Rinsing with diphoterine (126 workers) | Rinsing with water or no treatment (140 workers) | 0.75–38% | Time to return to work (days) |
| Huang, 2020 [12] | Taiwan, July 2010 to October 2017 | Retrospective observational study of records from the Taiwan Poison Control Center | 29 patients aged 22–48 years (male:female, 24:5) | TMAH * splashes of the skin | Rinsing with diphoterine | Rinsing with a decontaminant other than diphoterine | <1%—"Nearly entire body" | Depth of chemical burn (burn degree) |
| Kulkarni, 2018 [8] | India, September 2015 to November 2016 (14 months) | Prospective observational study of patients from The Ashirwad Clinic, Boisar | 65 patients aged 27–42 years (all male) | Both acid and alkali splashes of the skin | Rinsing with water plus diphoterine when admitted to hospital (nine patients) | Rinsing with water (56 patients) | 1–10% | Pain, VAS 0–10 (mean), time to return to work (days). |
| Nehles, 2006 [19] | Germany, 1994 to 1998 (4 years) | Retrospective observational study of records from a metallurgy | 24 workers aged 21–62 years (all male) | Both acid and alkali splashes | Rinsing with diphoterine | None | Head. Cheek. Thorax. Forearm. Face. Hand. Knee. Thorax, genitals, and thigh. | Complications |

**Table 1.** *Cont.*

| Author | Location and Study Period | Method | Study Group | Injury Type | Intervention | Control | Burn Area (TBSA [†]) | Assessment |
|---|---|---|---|---|---|---|---|---|
| Nogue, 2012 [14] | Spain, 2009 to 2011 (18 months) | Prospective observational study of patients from the Área de Urgencias Hospital Clínic, Barcelona | 9 patients aged 21–71 years (male:female, 5:4) | Both acid and alkali splashes | Rinsing with diphoterine (four patients) | Rinsing with water or soap and water (five patients) | Data not available | Pain, and complications |
| Škarja, 2014 [15] | Slovenia, 2014 | Randomized controlled trial, blinded | 36 firefighters; mean age of 38 years (all male) (no collected data for 1 one firefighter) | Burn with 5 mL 15% NaOH on 25 cm$^2$ skin for 55 s | Rinsing with diphoterine for 2 min (13 workers) | Rinsing with water for 2 min or no treatment (14 and 8 workers, respectively) | 25 cm$^2$ skin on forearm | Pain, NRS 0–10 (mean $\pm$ SE) Measured at 55 s, and 2, 3, 15, 60, 120, 240, and 360 min. after exposure to NaOH |
| Zack-Williams, 2015 [16] | United Kingdom, January 2010 to September 2012 (32 months) | Retrospective observational study of patients from Queen Elizabeth Hospital, Birmingham, UK | 131 patients, mean age of 37.7 for the diphoterine group and mean age of 43.2 for the control group (male:female, 104:26) | Both acid and alkali splashes of the skin | Rinsing with diphoterine (47 patients) | Rinsing with water (84 patients) | Mean diphoterine group: 1.76% mean water group: 1.25% | Duration of hospitalization (days), and pH |

[□] chlorobenzylidene malononitrile; [*] TMAH, tetramethylammonium hydroxide; [†] TBSA, total body surface area.

One retrospective study [12] analyzed records of chemical burns with tetramethy-lammonium hydroxide (TMAH) and reported the burn degree. The study included 29 cases in total. Of these, 5 cases were treated with diphoterine and 24 cases were not. Both the concentration of TMAH and the exposed body surface area (BSA) varied across the cases. Few cases in the diphoterine group and the control group were exposed to the same concentration of TMAH at the same BSA. No differences in burn degree were observed between the groups (Table 2).

Four studies assessed pain as an outcome, and three of the studies found a more pronounced decrease in pain after using diphoterine for irrigating a chemical burn compared to no treatment, irrigation with water, or irrigation with water and soap [8,13,14], while an RCT [15] found no difference between the diphoterine group and the water group.

One prospective observational study [16] assessed the duration of hospitalization and found no significant difference in the median length of stay.

An RCT [13] and two observational studies [8,17] evaluated time to return to work. None of the studies found any differences between the groups.

Two studies found significant improvements in pH when using diphoterine compared to water [16] and when using diphoterine as soon as possible [18].

An observational study [14] found no adverse reactions following the use of diphoterine. A retrospective study [19] also did not find any adverse effects associated with the use of diphoterine.

None of the studies reported the need for surgery.

A total of eight of the included studies were peer-reviewed [8,12–14,16–19], while one of the included studies was non-peer-reviewed [15].

The risk of bias was evaluated as high and with some concerns in the RCTs, and the NOS scores ranged between three and six stars (Table 3).

A meta-analysis was not performed due to an insufficient number of studies reporting the same outcomes in the same way.

*GRADE Assessment*

The strength and quality of the body of evidence were evaluated as "very low" across all outcomes. Due to the very different methods used throughout the studies and the lack of meta-analysis, inconsistency was not assessed. The outcome regarding complications when using diphoterine was not assessed with GRADE since the two articles describing this did not have complications as an actual outcome of interest, but rather, they spent a few lines describing it. The GRADE assessment is presented in Table 4.

**Table 2.** Outcomes from studies assessing diphoterine compared with water or no treatment for chemical burns of the skin.

| Outcome | Study | Measurement Scale | Diphoterine | Control | Between Group Estimate |
|---|---|---|---|---|---|
| Depth of chemical burn | Huang, 2020 [12] | Burn degree | 25% TMAH *, 5% BSA [†]:<br>- one case with first- to second-degree burn<br>25% TMAH, ≤1% BSA:<br>- two cases with first-degree burn<br>2.38% TMAH, <1% BSA:<br>- one case with first-degree burn<br>20% diluted TMAH, 1% BSA:<br>- one case with first-degree burn | 25% TMAH, 2% BSA:<br>- one case with second-degree burn<br>25%TMAH, ≤1% BSA:<br>- two cases with first-degree burn<br>2.38% TMAH, ≤1% BSA:<br>- five cases with first-degree burn, three cases with no burn<br>2.38% TMAH, ≤2% BSA:<br>- one case with first-degree burn one case with no burn<br>2.38% TMAH, BSA N/A (forearm):<br>- one/case with first-degree burn<br>0.50% TMAH, BSA "Nearly entire body":<br>- two cases with no burn<br>TMAH N/A, ≤1% BSA:<br>- five cases with first-degree burn one case with no burn<br>3% TMAH, BSA N/A (both forearms):<br>- one case with no burn<br>1–3% TMAH, <1% BSA:<br>- one case with first-degree burn | |
| Pain | Brvar, 2016 [13] | NRS 0–10, mean (95% CI) | Postexposure:<br>Inside CS [◻] cloud: 9.1 (9.1–9.1)<br>Residual pain at checkpoint 1.4 (1.3–1.4) | Inside CS cloud 9.7 (9.7–9.7)<br>Residual pain at checkpoint 2.3 (2.3–2.3) | |
| | Škarja, 2014 [15] | NRS 0–10, mean (SE) | 55 s: 11 had pain, NRS = 1.7 (±0.6)<br>2 min: one had pain, NRS = 1. Afterwards, no one had pain. | Water group:<br>55 s: 11 had pain, NRS = 2.3 (±0.9).<br>2 min: three had pain, NRS = 1.3 (±0.6). Afterwards, no one had pain.<br>No treatment group:<br>55 s: eight had pain, NRS = 2.1 (±1.0).<br>2 min: seven had pain, NRS = 2.3 (±1.0).<br>3 min: seven had pain, NRS = 2.0 (±1.2).<br>15 min: six had pain, NRS = 1.8 (±0.8).<br>60 min: five had pain, NRS = 1.2 (±0.4).<br>120 min: one had pain, NRS = 1. Afterwards, no one had pain | Between water group and diphoterine group at 2 min:<br>$p = 0.32$ |

**Table 2.** *Cont.*

| Outcome | Study | Measurement Scale | Diphoterine | Control | Between Group Estimate |
|---|---|---|---|---|---|
| | Kulkarni, 2018 [8] | VAS 0–10, mean | Before irrigation VAS = 7.0 After irrigation VAS = 3.1 Reduction: 3.9 | Before irrigation VAS = 5.7 After irrigation VAS = 4.1 Reduction: 1.6 | $p < 0.001$ |
| | Nogue, 2012 [14] | Descriptive | Three cases with "Rapid improvement of local symptoms" One case with "No effect on local symptoms" | Three cases with "Local symptoms better" Two cases with "No effect on local symptoms" | |
| Duration of hospitalization | Zack-Williams, 2015 [16] | Days, median | 1.75 days | 1.58 days | $p = 0.80$ |
| Time to return to work | Brvar, 2016 [13] | Min:sec (95% CI) | Preexposure: 1:26 (1:24–1:28) Postexposure: 2:30 (2:26–2:34) | CS only: 2:28 (2:26–2:30) | |
| | Kulkarni, 2018 [8] | Days (mean) | 2–7 days (4.67) | 3–20.2 days (16.75) | $p = 0.14$ |
| | Donoghue, 2010 [17] | Days | 1 restricted workday case 0 lost workday cases | 0 restricted workday cases. 0 lost workday cases | |
| Need for surgery | - | - | - | - | - |
| pH | Zack-Williams, 2015 [16] | pH, mean | Pre-irrigation pH = 8.07 pH change after treatment = 1.076 | Pre-irrigation pH = 7.77 pH change after treatment = 0.4 | $p < 0.05 / p = 0.000$ |
| | Cavallini, 2010 [18] | pH, mean | pH before treatment = 4.88 pH after glycolic acid = 0.7 pH after correction = 4.03 pH increase = 3.33 | pH before treatment = 4.88 pH after glycolic acid = 0.7 pH after correction = 3.4 pH increase = 2.7 | Increase, mean $p < 0.001$ |
| Complications | Nogue, 2012 [14] Nehles, 2006 [19] | | No ARs [§] No adverse effects | No ARs No adverse effects | |

* TMAH: tetramethylammonium hydroxide; [†] BSA: body surface area; [¤] chlorobenzylidene malononitrile; [§] adverse reaction.

**Table 3.** Risk-of-bias assessment for studies assessing diphoterine compared with water or no treatment for chemical burns of the skin.

| Risk of Bias in Randomized Trials, Using the Cochrane Collaboration's Tool for Assessing Risk of Bias in Randomized Trials | | | | | | |
|---|---|---|---|---|---|---|
| | Randomization Process | Deviations from the Intended Interventions | Missing Outcome Data | Measurement of the Outcome | Selection of the Reported Result | Overall Risk-of-Bias Judgement |
| Bvrar, 2016 [13] | High risk | Some concerns | Low risk | High risk | Some concerns | High risk of bias |
| Škarja, 2014 [15] | Some concerns | Some concerns | Low risk | Low risk | Some concerns | Some concerns |

| Risk of bias in non-randomized studies, using the Newcastle Ottawa Scale | | | | |
|---|---|---|---|---|
| | Selection (maximum of four) | Comparability (maximum of two) | Outcome (maximum of three) | Total |
| Cavallini, 2010 [18] | ★★ | ★★ | ★★ | 6 |
| Donoghue, 2010 [17] | ★★ | - | ★ | 3 |
| Huang, 2020 [12] | ★★★ | ★ | ★★ | 6 |
| Kulkarni, 2018 [8] | ★★ | ★ | ★ | 4 |
| Nehles, 2006 [19] | ★ | ★ | ★ | 3 |
| Nogue, 2012 [14] | ★★★ | - | ★★ | 5 |
| Zack-Williams, 2015 [16] | ★★★ | - | ★ | 4 |

Zero stars representing a high risk of bias, nine stars representing a low risk of bias.

**Table 4.** GRADE assessment. Author: Felicia Dinesen. Question: Diphoterine compared to water or no treatment for chemical burns of the skin. Bibliography: [8,12–19].

| Certainty Assessment | | | | | | | No. of Patients | | Effect | | | |
|---|---|---|---|---|---|---|---|---|---|---|---|---|
| No. of Studies | Study Design | Risk of Bias | Inconsistency | Indirectness | Imprecision | Other Considerations | Diphoterine | Water or No Treatment | Relative (95% CI) | Absolute (95% CI) | Certainty | Importance |
| **Depth of chemical burn** | | | | | | | | | | | | |
| 1 | Observational study | Serious [a] | | Serious [b] | Serious [c] | | 4 | 11 | - | - | ⊕○○○ VERY LOW | CRITICAL |
| **Pain** | | | | | | | | | | | | |
| 4 | Randomized trials and observational studies | Serious [d] | | Not serious | Serious [e] | | 34 | 89 | - | - | ⊕○○○ VERY LOW | CRITICAL |
| **Duration of hospitalization** | | | | | | | | | | | | |
| 1 | Observational study | Serious [f] | | Not serious | Serious [g] | | 47 | 84 | - | - | ⊕○○○ VERY LOW | IMPORTANT |
| **Time to return to work** | | | | | | | | | | | | |
| 3 | Randomised trial and observational studies | Serious [h] | | Not serious | Serious [i] | | 143 | 202 | - | - | ⊕○○○ VERY LOW | IMPORTANT |
| **pH** | | | | | | | | | | | | |
| 2 | Controlled trial and observational study | Not serious [f] | | Not serious | Serious [g] | | 72 | 109 | - | - | ⊕○○○ VERY LOW | IMPORTANT |

CI: Confidence interval. Explanations: [a] the study has no statement about follow-up and was adjusted only for type of chemical but not for age. [b] The study did not compare the depths of the chemical burns in the two groups. [c] Imprecise due to lack of comparison between groups and qualitative description of outcome. [d] RCTs performed on small, selected groups; one observational study was adjusted for age, and the other did not adjust for any of the confounding risk factors. [e] Most studies had few participants, and one study assessed the outcome with a qualitative description. [f] The observational study did not adjust for any of the confounding risk factors (age, additional treatment, and different chemical agents). [g] No CI reported. [h] RCT non-blinded and performed on a small, selected group; one observational study was adjusted for age, and the other did not adjust for any of the confounding risk factors. [i] Due to different methods and a lack of CI.

## 4. Discussion

Only one observational study reported on our primary outcome of depth of the chemical burn. In that study, the concentration of the burning agents, TMAH, and the burned BSAs varied across the cases. Only a few cases were exposed to the same concentrations of TMAH with the same BSAs, and no difference in the depth of the burns was observed. However, the main aim of that study was to describe the relationship between the extent of TMAH exposure and systemic toxicity, and no statistical analyses were performed regarding the depth of the chemical burns [12].

Diphoterine appears to be associated with less pain than rinsing with water, rinsing with water and soap, or receiving no treatment.

This systematic review was conducted following the Preferred Reporting Items for Systematic Reviews and Meta-Analysis (PRISMA) guidelines, which should ensure a systematic approach as well as the transparency of the methods used [9]. Furthermore, the protocol was written and registered [10] before initiating the review process to increase the transparency.

The limitations for this review must be acknowledged. The use of Google Scholar as a database was chosen to expand the possibility of finding relevant studies, but as a database, it has certain limitations. It is not possible to perform a search with as many details in the same way that is possible in other, more recognized databases, and neither is it possible to know the extent and correctness of the search. Furthermore, it is not possible to save the search and compare it with a later, similar search. Other limitations include the high heterogeneity of the data in the included studies, as well as the high risk of bias. Additionally, there is a lack of the consistent reporting of relevant data, and some studies did not report the burned total body surface area. Moreover, we chose to include a study with no comparator in order to describe complications associated with the use of diphoterine. Diphoterine is looked upon as a medical device and not as a drug, and this may lead to fewer studies on complications.

Both peer-reviewed and non-peer-reviewed studies were included in this systematic review. The peer-review process should ensure that the quality has been assessed by independent experts, and readers should be more critical when reading a non-peer-reviewed paper. It can, however, be reasonable to include such papers if few studies have been conducted, and one might argue that the processes of risk-of-bias assessment and quality assessment can serve as alternatives when evaluating a study. Both assessments are completed using checklists to support objectivity when evaluating the quality of a study.

Certain knowledge gaps exist regarding evidence on the use of diphoterine. In the included studies, diphoterine was used in different settings, and few reported on its use in a hospital setting. Furthermore, it is not possible to evaluate whether there is a lower limit for the size of the burned area to determine that diphoterine is superior to water.

Other reviews found that diphoterine was beneficial for chemical burns [7,20,21], but most were based on less clinically relevant study designs, including in vitro studies, and some studies were focused on subject's eyes, and some did not include a control group. Cautious interpretation is warranted when translating from pre-clinical studies, as acknowledged by others [7]. Animal skin differs from human skin, and only human studies were included in this review.

One review focused on human studies and concluded that diphoterine was safe and effective in promoting healing and pain relief [22]. In that review, no comparator was needed for the studies to be included. We think it is of high importance to include a comparator when evaluating a new treatment, since any new treatment should be implemented based on a better outcome than treatment with the already-existing treatment.

Based on our findings, the possible benefits of diphoterine may include a reduction in pain and neutralization after a chemical burn. The included studies did not quantify the intervention effect, and this makes it impossible to evaluate whether the difference is of clinical interest. One may argue that neutralization is a less clinically relevant outcome; nevertheless, neutralization might limit the extent of the damage from a chemical splash.

No significant differences in the duration of hospitalization or time to return to work were found. It is though worth mentioning that, even though Kulkarni [8] found no significant difference between the groups in terms of time to return to work, one might argue that there could be an economic interest in returning to work as fast as possible. Another possible benefit could be the reduced rinsing time, which would lower the risk of hypothermia, but this was not assessed.

Based on a very low certainty of evidence, this systematic review reports no observed difference between diphoterine compared with water or no treatment on the depth of a chemical burn. Diphoterine appeared to be associated with less pain compared to water or no treatment, and diphoterine appeared to have a neutralizing effect on chemical burns.

Evidence in this fields is very limited and heterogenic. It would be beneficial for further studies to focus on patient relevant outcomes, such as depth of burn, pain, and time to return to work.

**Supplementary Materials:** The following supporting information can be downloaded at: https://www.mdpi.com/article/10.3390/ebj4010006/s1, File S1: Prisma 2020 Checklist. Reference [23] is cited in the supplementary materials.

**Author Contributions:** Conceptualization, F.D., M.R.V. and L.S.R.; methodology, F.D. and L.S.R.; validation, F.D. and P.P.; formal analysis, F.D., P.P., M.R.V. and L.S.R.; investigation, F.D. and P.P.; resources, F.D. and P.P.; data curation, F.D.; writing—original draft preparation, F.D.; writing—review and editing, F.D., P.P., M.R.V. and L.S.R.; visualization, F.D.; supervision, M.R.V. and L.S.R.; project administration, F.D. and L.S.R.; funding acquisition, L.S.R. All authors have read and agreed to the published version of the manuscript.

**Funding:** This work was supported by the Department of Anaesthesia, Centre of Head and Orthopaedics, Rigshospitalet, Denmark.

**Conflicts of Interest:** The authors declare no conflict of interest.

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
