# Peer review of "Diphoterine for Chemical Burns of the Skin: A Systematic Review"

_2673-1991, doi:10.3390/ebj4010006_

Round 1

Reviewer 1 Report

hi

I can say that I like your work.

However, you excluded from the study the studies in which chemical burns were treated, especially with soap and water.

Publications with chemical adhesives, cement, pharmaceuticals, acetone, different acid variants are missing.

In the publication you focused only on the Diphoterine treatment.

sample publications: 1- Karahan, Z. A., & Kılıç, E. T. (2019). Rare chemical burns: first response, early hospitalization and first treatment:

2- Karahan, Z. A. (2019). Rare chemical burns: review of the literature.

etc.

Reviewer 2 Report

The authors have published a systematic review on an important topic, chemical burns.

In my opinion, however, this article in its current form is not suitable for publication. It includes an article (Nehles et al. 2006) that should not have been included based on an exclusion criterion (no comparison with water/no treatment). Also, the explanation in the results section regarding the inclusion process does not match Figure 1. Since the design of the study already contains errors, I believe that the authors should redo their systematic review before resubmitting the article.

I also included some other recommendations in each of the sections:

Introduction

·         Line 33-34: “Chemical burns differ from thermal burns in the way the damage is spreading, and treatment is therefore different as well.” The authors should explain this difference more into detail for the readers.

·         Line 34: Neutralization creates exothermic reaction, I would change this sentence to "quick elimination of the residual chemical product on the skin is key"

·         Line 45-46: this sentence should have a reference. Moreover, I recommend to add an explanation on Previn, otherwise this will not be clear for the readers in the materials and methods section.

·         Line 48: humans

Materials and methods

·         Line 59: systematic

·         Line 66: in my opinion, artificial human skin is not the human population that the authors aimed to investigate according to their PICO

·         Identify the authors who did the literature research, screening, … by their initials.

·         Line 70: Nehles et al is paper (case series?) without comparator and should have been excluded.

·         Line 83: as mentioned before: explain Previn in the introductions section. Otherwise this will not be clear for the readers.

·         Line 94: why did the authors not use a third author to obtain consensus as is done frequently?

·         Line 117: same: why did the authors not use a third author to obtain consensus as is done frequently?

Results

·         The results section does not match figure 1: e.g. in the table duplicates are removed before screening while in the text duplicates are removed from the 599 identified records. The authors should rewrite this for clarity.

·         Line 140: references of studies included.

·         Tables 1 and 2:

o   Add numbered references as this will make interpretation of the results easier for the readers.

o   Explain AR and CS for the readers

·         Line 8-10: “Few cases were comparable between the Diphoterine group and the control group regarding TMAH and BSA. When looking at these the burn degree was similar in the two groups.” The authors conclude this based on their observation. They should elaborate on this, because even the authors from the original article did not conclude this due to the differences in BSA and concentration of TMAH.

·         Line 17-18: “An RCT[9] and two observational studies[10,14] evaluated time to returning to work. None found any significant difference between groups.” Significant can only be used in case of statistical analyses. This is only done in one paper.

Discussion

·         Line 59-60: “Only one observational study reported on our primary outcome of depth of the chemical burn, and there was no difference in the depth of a TMAH burn between Di-60 photerine and other irrigation.” The authors cannot make this conclusion based on their observation and neither did the authors of the original paper.

·         Line 65: Meta-Analysis

·         Line 66-67:  references with [ ] instead of ( ).

Reviewer 3 Report

This is a well conducted and written manuscript.

 Abstract

18: please add the total number of studies found in your search

19: I would suggest to use ‘of which’ instead of ‘and’ (at the end of the line)

24: I would suggest to use ‘did appear’ instead of ‘appeared’

Introduction

Page 1

32: seems to increase: does it, or doesn’t it, or is the evidence inconclusive?

37: I would use ‘soon’ instead of ‘fast’, but please check with a native speaker.

38: on hand instead of by hand

Page 9

20: I would use ‘soon’ instead of ‘fast’, but please check with a native speaker.

Reviewer 4 Report

Manuscript: Diphoterine for chemical burns of the skin; a systematic review

The authors have done a thorough systematic review on an important issue. The main drawback is the small number of publications that were adequate to include and the low grade of quality among these studies, limitations that the authors have already addressed in the discussion.

The manuscript reads well. The method is clearly described, the results section is well structured, the tables are informative, and the discussion is focused on relevant topics.

Minor Comments:

It is interesting that not peer reviewed papers were included. Please clarify in the manuscript for which of the publications this was the case.

Also, please consider to add your thoughts in the discussion regarding the value of not peer reviewed papers. It would be interesting to read your thoughts on whether the quality assessment under the protocol of a systematic review can be as good (or even better in some aspects) as the peer review process.

Please consider to use the English study title in the Reference list for ref #12, using [square brackets] to indicate other original language.

Round 2

Reviewer 2 Report

The reviewer thanks the authors for answering the comments.

However, I still disagree with some key points of the article. Therefore, I believe that the article is still not suitable for publication.

Abstract

·         Line 14-16: strange sentence now.

Introduction

·         Line 32: it should be “… to 7.9% OF total burn injuries…”

Materials and methods

·         Line 71-73 and 78-80: in my strong opinion, the authors cannot say that We identified studies recruiting humans exposed to a chemical splash on the skin, being acid, or alkali, liquid or gas (population), that was rinsed with Diphoterine (intervention) or water or that received no treatment (comparator).” and “We excluded studies on Diphoterine without a comparator.” and than say that “studies on complications associated with Diphoterine were included even if no comparator was part of the study.” to include a study deemed interesting to the authors. They only concluded that this study “also did not find any adverse effects associated with the use of Diphoterine.” Additionally, the authors describe later that “since the two articles describing this, did not have complications as an actual outcome of interest, but rather, they spent a few lines on describing it.” One can ask why it is than so important to try to include that one paper although they tried to explain it in the discussions section.

·         Line 71: my opinion stays the same: artificial human skin is not the human population that the authors aimed to investigate according to their PICO. I do not agree with the answer of the authors that “artificial human skin is an acceptable model”. It should be removed from the search even if there were no studies identified and included in this systematic review.

Results

·         Tables 1 and 2: as far as the reviewer can see, the authors did not add reference numbers to the papers in both tables (point 8). This although the authors declared that they did this.

·         Line 10-11: I still not agree with the conclusion of the authors. “When looking at these the burn degree was similar in the two groups.”. As already mentioned, this is a conclusion the authors of this systematic review made and not the authors of the original paper. No statistical work was performed to proof this.

Discussion

·         Lin 62-67: “Only one observational study reported on our primary outcome of depth of the chemical burn., and there was no difference in the depth of a TMAH burn between Diphoterine and other irrigation. In that study the concentration of the burning agents, TMAH, and the burned BSAs varied across the cases. Only a few cases were exposed to the same concentrations of TMAH with the same BSAs. No differences between the groups were found for depth of burns[12].” Same as mentioned in the results section. The authors of the original article did not (statistically) compare both groups, so the authors of this systematic review cannot make this conclusion. The same holds for their conclusion in line 122-124.
